# Evaluation of Cysteine Metabolism in the Rat Liver and Kidney Following Intravenous Cocaine Administration and Abstinence

**DOI:** 10.3390/antiox10010074

**Published:** 2021-01-08

**Authors:** Danuta Kowalczyk-Pachel, Małgorzata Iciek, Anna Bilska-Wilkosz, Magdalena Górny, Joanna Jastrzębska, Kinga Kamińska, Paulina Dudzik, Małgorzata Filip, Elżbieta Lorenc-Koci

**Affiliations:** 1The Chair of Medical Biochemistry, Medical College, Jagiellonian University, 7, Kopernika St., PL-31-034 Kraków, Poland; d_kowalczyk85@o2.pl (D.K.-P.); malgorzata.iciek@uj.edu.pl (M.I.); mbbilska@cyf-kr.edu.pl (A.B.-W.); mbgorny@cyf-kr.edu.pl (M.G.); paulina.dudzik@uj.edu.pl (P.D.); 2Maj Institute of Pharmacology, Polish Academy of Sciences, 12, Smętna St., PL-31-343 Kraków, Poland; czyzyk@if-pan.krakow.pl (J.J.); k.kamin@if-pan.krakow.pl (K.K.); mal.filip@if-pan.krakow.pl (M.F.)

**Keywords:** cocaine, reactive sulfur species, self-administration, thiols, yoked procedure

## Abstract

Many toxic effects of cocaine are attributed to reactive oxygen species (ROS) generated during its metabolism. Recently, it has been suggested that the biological action of ROS is often confused with endogenously generated reactive sulfur species (RSS). The aim of this study was to evaluate the impact of cocaine on thiols and RSS in the rat liver and kidney in the drug self-administration (SA) paradigm and the cocaine yoked delivery model (YC) followed by drug abstinence with extinction training. The level of thiols as well as RSS formed during anaerobic metabolism of cysteine and sulfate were assayed. In addition, the activity of enzymes involved in RSS formation and glutathione metabolism were determined. In the liver, following direct cocaine administration (SA and YC), the RSS levels decreased, while in the kidneys, cocaine increased the RSS contents in both groups. These changes were maintained in these tissues during drug abstinence. The level of sulfates was changed by cocaine only in the liver. In the kidney, cocaine shifted cysteine metabolism towards an anaerobic pathway. Our study demonstrates for the first time the changes in cysteine metabolism and thiol levels in the liver and kidney of rats after cocaine self-administration and abstinence.

## 1. Introduction

Cocaine, an alkaloid derived from the leaves of the plant Erytroxylon coca growing in South America, is a potent psychoactive substance. It is a stimulant of the sympathetic nervous system acting by inhibition of dopamine (DA) reuptake from the extracellular space, and its transmission to brain structures is associated with pleasure. This buildup causes intense feelings of energy and pleasure [1,2]. Powerful stimulation of the central nervous system by cocaine results in its high addictive potential; hence, cocaine is one of the most addictive and widely use drugs [3].

Many studies have shown that cocaine causes irreversible structural changes in the brain; however, damages to other tissues, such as the heart, lung, liver, and kidneys, have also been documented [1,4]. In the liver, most of the direct toxic effects of cocaine are related to its cytochrome P450 (CYP450)-mediated metabolism to norcocaine, a compound with potent pro-oxidant properties. Apart from norcocaine, other cocaine metabolites are also capable of producing reactive oxygen species (ROS) and thus of inducing oxidative stress [5]. Due to the highest activity of CYP450 in the liver, cocaine toxicity should be mainly restricted to this organ, but it has also been observed in other tissues. Therefore, it has been suggested that mechanisms other than those induced by reactive metabolites and ROS may contribute to the multi-organ toxicity of cocaine [1,6].

Recently, some authors have suggested that the biological action of ROS is often confused with endogenously generated reactive sulfur species (RSS) [7]). RSSs are a group of redox active molecules which are formed in vivo during cysteine (Cys) transformations and mainly include hydrogen sulfide (H_2_S) and products of its oxidation: inorganic polysulfides (H_2_S_n_) as well as hydropersulfides (RSSH) [8,9]. RSSs are omnipresent and play regulatory roles in biological systems. They are highly reactive and are considered the leading players in redox control, perhaps being more important than ROS [10]. RSS and ROS are chemically similar, and both transmit signals via oxidation reactions with Cys protein sulfur, but RSS appears to be more effective [7]. It seems that many of the biological attributes previously assigned to ROS may in fact result from the action of RSS, especially since these species are often indistinguishable by analytical methods [11].

Since most of the direct toxic effects of cocaine are attributed to ROS and oxidative stress, in this study, we decided to focus on the effects of cocaine on thiols and RSS. Protein and non-protein thiols play an important role in the reducing power of the cells. Glutathione (GSH) is the most widespread low-molecular-weight antioxidant in the liver and kidney. Cys is not only necessary for GSH synthesis but also a source of RSS produced during anaerobic metabolism. RSS include sulfane sulfur, bound sulfane sulfur (mainly persulfides), and H_2_S. The enzymes involved in the formation or transport of these RSSs comprise cystathionine γ-lyase (CSE), 3-mercaptopyruvate sulfurtransferase (MST), and rhodanese (TST) [12,13,14]. On the other hand, aerobic Cys metabolism leads to taurine and sulfate (Figure 1).

The effect of cocaine on thiol homeostasis in peripheral tissues is poorly understood, so far. A significant reduction in the total thiol content was detected in the human plasma of cocaine users [15]. Our previous study showed that cocaine, dependent on the method of administration, induced significant changes in the homeostasis of thiol amino acids and sulfane sulfur in rat plasma [16]. We have also recently conducted studies on peripheral tissues of rats (liver and kidneys) after acute and subchronic intraperitoneal administration of cocaine [17]. That study revealed that a single cocaine injection, just like repeatedly administered cocaine, enhanced the pool of sulfane sulfur in rat kidneys [17]. The latter results clearly suggest that, in the kidney, cocaine shifts Cys metabolism towards the anaerobic route, yielding compounds with sulfane sulfur, which possesses regulatory and antioxidant properties. It has also been documented that *N*-acetylcysteine (NAC), which is a precursor of both Cys and sulfane sulfur, produced a significant decline in cocaine-induced reinstatement in an animal model [18,19,20,21].

The data presented above led us to hypothesize that the changes in RSS may be implicated in the mechanism of cocaine addiction in the rat model of cocaine self-administration (SA). In this model, rats are taught to press the lever that leads to intravenous injection of the drug. They are rewarded in this way, and lever pressing is positively reinforced. The amount and frequency of cocaine injection given to the animal indicated the intensity of addiction. The SA procedure was extended by the “yoked” procedure (YC) to separate the motivational and pharmacological effects of the drug. Each cocaine self-administering animal was coupled to another rat passively receiving the drug (the same amount and frequency of injection). In the current work, hepatic and renal sulfur metabolism was also tested after 10 days of cocaine abstinence with extinction training after the SA and YC procedures.

Therefore, the aim of the present study was to investigate the effect of cocaine in the self-administration model with the yoked procedure and after 10-day abstinence with extinction training on the levels of sulfhydryl (–SH) groups: total (TSH), non-protein (NPSH), and protein (PSH); on anaerobic Cys metabolism leading to the formation of total sulfane sulfur, bound sulfane sulfur, and H_2_S; and on the level of sulfate, which is the product of aerobic Cys transformation. The activities and expression of the enzymes involved in sulfane sulfur formation and transport (CSE, MST, and TST) and the enzymes related to GSH metabolism (γ-glutamyltranspeptidase (γ-GT) and glutathione S-transferase (GST)) were also measured.

We hope that this broad spectrum of research and the use of the cocaine self-administration and drug abstinence with extinction training models will contribute to expanding our knowledge of the new mechanisms of cocaine action related to Cys metabolism in peripheral tissues.

## 2. Materials and Methods

### 2.1. Ethics Statement

The experiments were carried out in compliance with the Act on Experiments on Animals of 21 January 2005 reapproved on 15 January 2015 (published in the *Journal of Laws* no. 23/2015 item 266, Poland) and according to the Directive of the European Parliament and of the Council of Europe 2010/63/EU of 22 September 2010 on the protection of animals used for scientific purposes. They were also approved by the local ethics committee at the Institute of Pharmacology, Polish Academy of Sciences. All efforts were made to minimize the number and suffering of animals.

### 2.2. Animals

Male Wistar rats (Charles River Laboratories, Cologne, Germany) weighing 280–300 g at the beginning of the experiment were housed in standard home cages (4 animals/cage) in a temperature-controlled room at 20 ± 1 °C and at 40–50% humidity under a 12-h light-dark cycle (lights on at 06:00 a.m). Animals had free access to food (Labofeed pellets) and water except the initial 2 lever press-training and retraining sessions, during which rats were maintained on limited access to water (see below). All experiments were conducted during the light phase of the light-dark cycle (between 08:00–15:00).

### 2.3. Cocaine Self-Administration and Extinction Training with “Yoked” Procedure

All animals used in these studies underwent the same training and surgical procedures (Figure 2). First, all rats were water-deprived for 18 h and then trained for 3–4 days in standard operant cages (Med-Associates Inc., Fairfax, VT, USA) to press the lever for 2 h daily for water reinforcement under a fixed ratio (FR) at schedule 1. In the case of cocaine self-administration, on the fourth day of lever press training, the number of responses required to produce reinforcement increased to schedule 5, which means that every 5 lever presses on the “active” lever resulted in delivery of one portion of water [22]. After the training sessions, rats were allowed 10 min of free access to water in their home cages [22]. Two days after the training sessions, rats were implanted with a silastic catheter into the external jugular vein, as described previously in detail [22]. After surgery, rats were allowed 7 days to recover. Cocaine hydrochloride (National Institute on Drug Abuse, RTI International, Research Triangle Park, NC, USA) was dissolved in sterile 0.9% NaCl and given i.v. (0.1 mL/infusion). The cocaine self-administration procedure was preceded by a 2-day lever-press retraining session. Then, the animals were placed in standard operant chambers (mentioned above) and allowed to self-administer cocaine (0.5 mg/kg/infusion; 0.1 mL) during 2-h daily sessions under the fixed ratio schedule 5 of reinforcement for 14 days. A tone (2000 Hz; 15 dB above ambient sound levels) and illumination of the stimulus light directly above the “active” lever were presented for 5 s, concurrent with a successful response for cocaine; following each injection, there was a 20-s time-out period during which responses were recorded but had no programmed consequences. Response on the “inactive” lever never resulted in cocaine delivery (Figure 2).

One group of rats was sacrificed by decapitation immediately after the last 2-h cocaine self-administration session following a 14-day series of cocaine self-administration, while another group underwent a 10-day abstinence with extinction training (Figure 2). During extinction training, the animals had 2-hr daily training sessions with no delivery of cocaine or presentation of the conditioned stimulus. Rats were sacrificed by decapitation immediately following the session (Figure 2), while their livers and kidneys were used for further biochemical assays.

To generate a proper control group, a yoked “triad” control procedure described by Frankowska et al. [22] was applied. Animals were tested simultaneously in groups of three: rats actively self-administering cocaine (SA group) were paired with rats passively receiving cocaine (YC; yoked cocaine) and saline (YS, yoked saline, control group). Yoked groups were sacrificed by decapitation at the same time as rats self-administered cocaine or those that underwent extinction training.

### 2.4. Chemicals

1-chloro-2,4-dinitrobenzene (CDNB), dithiothreitol (DTT), 5,5′-dithio-bis-2-nitrobenzoic acid (DTNB), gelatin, glycine-glycine (Gly-Gly), glutathione reduced form (GSH), lactic dehydrogenase (LDH), l-glutamyl-3-carboxy-4-nitroanilide, 3-mercaptopyruvate (3-MP), 3-methyl-2-benzothiazolinone hydrazine hydrochloride monohydrate (MBTH), NADPH, *N*-(1-naphthyl)-ethylene-diamine hydrochloride, *N*-ethylmaleimide (NEM), β-nicotinamide adenine dinucleotide reduced form (NADH), 2-nitro-5-thiobenzoic acid (TNB), p-phenylenediamine, potassium cyanide (KCN), potassium thiocyanate (KSCN), pyridoxal 5′-phosphate monohydrate (PLP), pyruvate, sodium hydrosulfide (NaHS), sodium potassium tartrate, sodium salt l-homoserine, sodium thiosulfate (Na_2_S_2_O_3_), sulfanilamide, trichloroacetic acid (TCA), thionine, and zinc acetate were provided by Sigma-Aldrich Chemical Company (St. Louis, MO, USA). Acetic acid, ammonia (NH_3_), barium chloride (BaCl_2_), copper sulfate (CuSO_4_), Folin reagent, formaldehyde, ferric chloride (FeCl_3_), hydrochloric acid (HCl), iron nitrate (Fe(NO_3_)_3_), magnesium chloride (MgCl_2_), nitric acid (HNO_3_), potassium dihydrogen phosphate (KH_2_PO_4_), perchloric acid (HClO_4_), sodium carbonate (Na_2_CO_3_), sodium hydroxide (NaOH), sodium sulfite (Na_2_SO_3_), thiosulfate, and all other reagents were obtained from the Polish Chemical Reagent Company (P.O.Ch, Gliwice, Poland). Anti-CSE, anti-MST, and anti-TST antibodies were from Santa Cruz Biotechnology (Dallas, TX, USA), whereas anti-Hsp90 and HRP-conjugated secondary antibody as well as LumiGLO reagent were purchased from Cell Signaling Technology (Danvers, MA, USA; Chicago, IL, USA).

### 2.5. Preparation of Tissue Homogenates

All experimental procedures involved in the preparation of tissue homogenates were carried out at 4 °C. The frozen tissue samples of liver and kidney were weighted and homogenized using an IKA-ULTRATURRAX T8 homogenizer (1 g of the tissue in 4 mL of 0.1 M phosphate buffer, pH 7.4). The obtained homogenates were used for biochemical assays.

### 2.6. Biochemical Assays

Determination of the non-protein, protein, and total –SH groups (NPSH, PSH, and TSH, respectively) was carried out by the Ellman’s reaction [23]. The PSH level was calculated for each tissue as a difference between TSH and NPSH.

The total level of sulfane sulfur was determined by the method of Wood [24], while the level of bound sulfane sulfur was determined by the modified method of Ogasawara et al. [25]. This pool of sulfane sulfur includes mainly persulfides and polysulfides, which release H_2_S by reduction with dithiothreitol (DTT). The level of H_2_S was determined using a modified method of Shen et al. [26] with fluorometric detection.

The level of sulfate was estimated according to the manufacturer’s instructions for the Sulfate assay kit (Sigma), in which inorganic sulfate is precipitated by a reaction with barium sulfate. Gelatin solution was used instead of polyethylene glycol for stabilization turbidity.

The activity of CSE was assayed by the modified method of Matsuo and Greenberg [27]. CSE catalyzes the transformation of l-homoserine to α-ketobutyric acid, which is then assayed according to the method of Soda [28].

The activity of MST was determined by the method of Valentine and Frankenfeld [29], while TST activity was assayed according to Sörbo’s method [30].

Estimation of the γ-GT activity was performed by the method of Orłowski and Meister [31], while GST activity was assayed as described previously [32].

The level of protein was assayed by the method of Lowry et al. [33].

All these procedures were previously described in detail [17].

### 2.7. Western Blot Analysis

Rat kidney tissue was homogenized in phosphate buffer, and protein concentration was determined by Lowry’s method [33]. Protein lysate (20 mg) was separated by SDS PAGE (10%) and blotted onto nitrocellulose membrane (Thermo Fisher Scientific, Waltham, MA, USA). Membranes were blocked with 5% milk in TBS-Tween over 1 h. The proteins of interest were detected by the following primary antibodies: anti-CSE (1:1000), anti-TST (1:1000), and anti-MST (1:1000) in 5% milk/TBS-Tween. The anti-Hsp90 (1:4000) antibody was used to check for equal loading. Hsp90 has a molecular weight of 90 kDa, and its location on the membrane allowed for the simultaneous analysis of all interested proteins (CSE, 45 kDa; MST, 33 kDa; and TST, 33–35 kDa).

All proteins of interest were detected using HRP-conjugated secondary anti-mouse antibody 1:2000 (CSE, TST, and MST) and 1:10,000 (Hsp90) in 5% milk/TBS-Tween. Proteins were visualized by the chemiluminescence method using the LumiGLO reagent. The chemiluminescence intensity was recorded with a Bio-Rad ChemiDoc^TM^ MP Imaging System (Bio-Rad, Hercules, CA, USA). Kidney tissue samples from 3 rats from each group (YS, SA, and YC) were chosen for Western Blot analysis. Densitometric analysis of the obtained bands was performed using ImageJ Image Processing and analysis software version 2.1.4.7 (National Institutes of Health, U.S. Department of Health and Human Servises, Washington, DC, USA). The bands for the proteins of interest (CSE, MST, and TST) were quantified relative to the band for the corresponding loading control (Hsp90).

### 2.8. Statistics

A one-way analysis of variance (ANOVA) followed (if significant) by Dunnett test was used for statistical analysis of biochemical data. A two-way ANOVA for repeated measures followed by Tukey test was used for comparisons of behavioral effects between yoked saline (YS), cocaine self-administered (SA), and yoked cocaine (YC) groups. A *p* value < 0.05 was considered statistically significant.

## 3. Results

### 3.1. Behavioral Studies

Rats self-administering cocaine showed stable lever-pressing rates during the last three self-administration days, with the mean number of cocaine infusions per day varying from 28 to 31. During 14 experimental sessions, animals received ca. 185 mg/kg of cocaine/rat. Rats pressed significantly (*p* < 0.001) more frequently on the “active” lever in comparison to the “inactive” lever from the 3rd to 14th experimental session (F(13,234) = 12.66) (Figure 3A), while those animals that underwent 10-day cocaine abstinence pressed significantly more frequently (*p* < 0.001) on the “active” lever than on the “inactive” lever from the 3rd to 18th experimental session (F(23,414) = 12.08) (Figure 3B). As shown in Figure 3B, during the last 3 days of cocaine abstinence, the total number of “active” lever presses differed by less than 10%.

The “yoked” cocaine animals received exactly the same amount of cocaine (185 mg/kg/rat) at the same time as the rats that had learned to self-administer cocaine. In the “yoked” cocaine and “yoked” saline groups, the difference between pressing the “active” and the “inactive” lever failed to reach significance (data not shown).

### 3.2. Effect of Self-Administration of Cocaine (SA) or Its Passive Infusions (YC) on the Concentrations of Sulfane Sulfur and Its Bound Fraction as Well as on H_2_S and Sulfates in the Rat Liver and Kidneys

The level of the whole pool of sulfane sulfur in the liver and kidney did not differ significantly between the studied groups (SA, YC, and YS), during self-administration and abstinence (Figure 4A,B and Figure 5A,B). However, the level of bound sulfane sulfur in the self-administration (SA) and “yoked” cocaine groups (YC) significantly decreased in the liver, while in the kidneys, it increased compared to the yoked saline (YS) control during cocaine self-administration and drug abstinence with extinction training (Figure 4C,D and Figure 5C,D).

As for the hepatic H_2_S, its level during self-administration significantly decreased only in the YC group (Figure 4E) and during drug abstinence only in the SA group compared to the YS control (Figure 4F). In contrast to the liver, the level of H_2_S in the kidneys increased significantly in the SA and YC groups during cocaine self-administration (Figure 5E) but only in the YC group during drug abstinence (Figure 5F).

As for hepatic sulfates in the presence of cocaine, their levels significantly increased in self-administering rats, while in the YC group, it clearly decreased vs. the YS control (Figure 4G). However, during cocaine abstinence with extinction training, sulfate levels decreased significantly only in the SA group, while in YC group, it almost returned to the control level (Figure 4H). Contrary to the liver, no significant differences were found in renal sulfate concentrations between the studied groups (SA, YC, and YS) during cocaine administration and drug abstinence with extinction training (Figure 5G,H).

### 3.3. Effect of Self-Administration of Cocaine (SA) or Its Passive Infusions (YC) on Enzymatic Activities of Sulfurtransferases (CSE, MST, and TST) in the Rat Liver and Kidneys

The activity of CSE in the rat liver was not significantly different in any of the cocaine treated groups (SA and YC) compared to the YS control in both the presence of the drug and cocaine abstinence with extinction training (Figure 6A,B). In contrast, in the kidneys, the CSE activity decreased significantly in SA and YC groups in the presence of cocaine (Figure 7A), while in drug abstinence with extinction training, it increased only in the YC group compared to the YS control (Figure 7B).

The activity of MST in the livers of rats from the SA and YC groups increased significantly in the presence of cocaine (Figure 6C), but in the kidneys in both these groups, it was clearly reduced compared to the YS control (Figure 7C). After 10 days of cocaine abstinence with extinction training, MST activity in the liver and kidneys of the SA and YC groups returned to the control level (Figure 6D and Figure 7D).

The activity of TST in the liver of SA and YC rats increased significantly in the presence of cocaine (Figure 6E) but did not differ in the kidneys in both groups compared to the YS control (Figure 7E). After 10 days of cocaine abstinence with extinction training, TST activity in the liver of SA and YC groups returned to the control level (Figure 6F), while in the kidneys, it decreased significantly in both groups compared to the YS control (Figure 7F).

### 3.4. Effect of Self-Administration of Cocaine (SA) or Its Passive Infusions (YC) on the Expression of CSE, MST, and TST Protein in the Rat Kidney

Since the levels of bound sulfane sulfur and H_2_S in the presence of cocaine and after 10 days of drug abstinence with extinction training were significantly increased in the kidneys of the SA and YC rats, in addition to the enzymatic activities of CSE, MST, and TST (Figure 7), protein expression of these enzymes was also measured only in this tissue (Figure 8). Western blot analysis showed that the level of CSE protein was significantly increased in the kidneys of SA rats (Figure 8A) while, in the YC group, it remained at the level of the YS control both in the presence of cocaine and after its abstinence (Figure 8A,B). As for MST, expression of this protein in the presence of cocaine tended to increase only in the SA group while, after 10 days of abstinence with extinction training, such a tendency was observed both in the SA and YC groups (Figure 8C,D). However, the most characteristic change was the significant increase in the level of TST protein in the studied groups (SA and YC) both in the presence of cocaine and after its abstinence with extinction training (Figure 8E,F).

### 3.5. Effects of Self-Administration of Cocaine (SA) or Its Passive Infusions (YC) on the Levels of Particular Fractions of –SH Groups in the Rat Liver and Kidneys

The levels of the individual fractions of the –SH groups (TSH, NPSH, and PSH) in the livers of the cocaine self-administering rats (SA) were almost the same as that in the YS control group (Figure 9A,C,E). Similar effects were seen in the liver of the YC group with the exception of the NPSH level, which decreased slightly but significantly compared to the YS group (Figure 9C). After cocaine abstinence with extinction training, the levels of TSH and PSH in the liver decreased significantly in both the SA and YC rats (Figure 9B,F), and only NPSH content (mainly GSH) in these groups remained at the level of the YS control (Figure 9D).

In contrast to the liver, in the kidneys of the SA group, the levels of TSH, NPSH, and PSH increased significantly compared to the YS group (Figure 10A,C,E). Slightly weaker but still significant increases in TSH and NPSH levels, with the exception of PSH, were also seen in the YC group (Figure 10A,C,E). After cocaine abstinence with extinction training, TSH and PSH levels were still clearly elevated in SA and YC groups (Figure 10B,F) and only the NPSH content decreased in the SA group, while in the YC group, it was maintained at the level of the YS control (Figure 10D).

### 3.6. Effects of Self-Administration of Cocaine (SA) or Its Passive Infusions (YC) on Enzymatic Activities of γ-GT and GST in the Rat Liver and Kidney

In the rat liver, the activity of γ-GT after the last cocaine dose increased significantly in SA and YC groups (Figure 11A), while in the kidneys of both groups, the activity of this enzyme decreased markedly compared to the YS control (Figure 12A). After cocaine abstinence with extinction training, the activity of γ-GT was still significantly increased in the liver of the YC group (Figure 11B), while in the kidneys, such an effect was only seen in the SA group (Figure 12B).

In the rat liver, GST activity also increased significantly after cocaine administration both in the SA and YC groups compared to the YS control (Figure 11C), while in the kidneys, it was clearly reduced but only in the YC group (Figure 12C). After cocaine abstinence with extinction training, in the rat liver, GST activity was significantly reduced only in YC group compared to the YS control (Figure 11D), while in the kidneys, a downward trend was observed in this enzyme activity in the YC group (Figure 12D).

## 4. Discussion

The present study shows for the first time changes in Cys metabolism in the liver and kidneys of rats induced by intravenous self-administration of cocaine. The rat model of cocaine self-administration is the best for monitoring the abovementioned processes because it most accurately reflects the conditions of human cocaine use. In addition, a group of rats self-administering cocaine was compared with a group of rats administered cocaine by an experimenter (yoked), which allows for distinguishing between motivational and pharmacological effects of cocaine. The biochemical effects of cocaine have most often been investigated during cocaine reinforcement, but we believe that drug abstinence with extinction training is equally interesting because changes occurring during this period may contribute to finding an effective therapy able to prevent relapse of addiction. Therefore, to comprehensively illustrate the changes in Cys metabolism following cocaine administration, both models described above were used in the present study. This is an extension of our earlier observation that cocaine alters Cys metabolism in the same tissues after single and subchronic intraperitoneal administration [17].

The results obtained in the present study indicate that, regardless of the mode of cocaine delivery, this drug affects thiol balance and that some of the biochemical parameters after cocaine abstinence with extinction training differ from those in its presence. In the studied models of cocaine administration, both in the liver and kidneys, no significant differences were observed in the total sulfane sulfur pool; however, interesting differences appeared in the concentrations of bound sulfane sulfur and H_2_S. Overall, in the liver of SA and YC rats during cocaine administration and following drug abstinence with extinction training, there was a decrease in the level of RSS, i.e., primarily bound sulfane sulfur (RSSH) and H_2_S (Figure 4), while in the kidney, in both these groups, cocaine increased the levels of the tested RSS compounds (Figure 5). On the other hand, in our previous study performed with both cocaine administration models (SA and YC), this drug significantly lowered the total pool of sulfane sulfur in the rat plasma [16].

Sulfane sulfur is formed mainly in tissues, especially in the liver, and then can be transported to the plasma in the form of albumin persulfides [12,14]. The observed decrease in the total pool of sulfane sulfur in plasma in the absence of changes in its content in the liver and kidneys in the current study may result from cocaine-induced inhibition of sulfane sulfur transport from hepatic cells to the plasma or from its greater use to compensate for oxidative stress [34]. In line with the latter explanation, in the present study, an increase in the activity of the MST and TST enzymes responsible for the formation of sulfane sulfur and its export, respectively, was observed in the liver from SA and YC groups during cocaine intake (Figure 6C,E). Furthermore, in the previous study, after cocaine abstinence with extinction training, in the SA group, the level of total sulfane sulfur in the plasma returned almost to the control value while, in the YC group, it increased significantly compared to the YS and SA groups [16]. In the present study, these changes in total plasma sulfane sulfur concentrations, observed during cocaine abstinence in our earlier research [16], were accompanied by small but significant decreases in the contents of bound sulfane sulfur and H_2_S in the liver of rats from SA group (Figure 4D,F), while in the YC group, significantly greater decreases in the content of bound sulfane sulfur (Figure 4D) were see than in the SA group. At the same time, in the SA and YC groups, the hepatic activity of MST and TST returned almost to the control value (Figure 6D,F). The results presented above seem to indicate that, in the liver of the SA and YC groups, the decreases in the concentration of bound sulfane sulfur during cocaine abstinence with extinction training may be due to the increased plasma sulfane sulfur requirement to compensate for the effects of oxidative stress.

In contrast to the liver, in the kidneys, cocaine had different effects on the levels of sulfane sulfur and H_2_S and on the activity of enzymes involved in their formation (CSE and MST). It is worth mentioning here that the liver and kidney differ fundamentally in efficiency of Cys metabolism. The liver is characterized by high levels of GSH, the major non-protein thiol, and low Cys. The kidneys, on the other hand, are characterized by a lower concentration of GSH than in the liver and the highest level of Cys among all body tissues [33,34,35,36]. As a consequence, anaerobic Cys transformations leading to RSS formation intensify in the kidney; hence, the highest levels of both total sulfane sulfur and its bound fraction were found in this organ [37].

In the present study, increases in the levels of bound sulfane sulfur and H_2_S were observed in the kidneys of SA and YC rats during cocaine intake (Figure 5C,E). Furthermore, after drug abstinence with extinction training, the levels of bound sulfane sulfur were still increased in the SA and YC groups (Figure 5D) but H_2_S level was enhanced only in the YC rats (Figure 5F). In our previous study, in the kidney of rats that received cocaine repeatedly by i.p., a significant increase in the level of total and bound sulfane sulfur and a decrease in the content of H_2_S were shown [17]. Interestingly, acute i.p. administration of cocaine resulted in a dramatic decrease in bound sulfane sulfur with a simultaneous increase in the total sulfane sulfur pool and no changes in the level of H_2_S in the kidneys [17]. This means that the route of administration (i.v. and i.p.) and the dose of the drug can affect the amount of free H_2_S. Furthermore, our results indicate that, in rat kidneys, both during cocaine intake and following drug abstinence with extinction procedure, Cys metabolism is shifted towards anaerobic pathway.

H_2_S at low physiological concentrations plays an important regulatory role, is an inorganic substrate for the mitochondrial respiratory chain, and thus serves as an electron donor for ATP generation [38]. However, in high concentrations, H_2_S is dangerous because it inhibits the mitochondrial respiratory chain by blocking the IV complex. Therefore, the amount of H_2_S produced in the body, especially in the kidneys, must be tightly controlled, as both deficiency and excess of this gaseous transmitter are harmful to the body [39]. It is thought that H_2_S is stored in the form of persulfides and polysulfides, which represent bound sulfane sulfur [40]. They are believed to be normal cellular components involved in redox signaling and in the protection of Cys residues in proteins and low molecular weight thiols from oxidative stress [41,42]. Hence, an increase in the level of bound sulfane sulfur in the kidney of SA and YC rats in our study may mean both a stronger protective and reducing power of the cells. In general, the increase in RSS levels may be due to the activation or increased protein expression of the enzymes involved in their synthesis. In our studies, the observed increase in the expression of sulfurtransferases, especially in the SA group, was accompanied by a decrease in the activity of these enzymes, which indicates a mechanism that limits excessive RSS production, possibly through posttranslational modifications of these proteins or other types of regulation (e.g., feedback). Moreover, an increased expression of sulfurtransferases after 10 days of cocaine abstinence with extinction training indicates that cocaine causes long-term adaptive changes at the transcriptional level. The involvement of sulfurtransferases in the effects of cocaine may be of great importance for elucidating the processes responsible for cocaine addiction, and therefore, this issue requires further detailed research.

Tissue sulfate concentrations, formed during aerobic metabolism of Cys, in the presence of cocaine and after 10 days of drug abstinence with extinction training changed only in the liver (Figure 4G,H). In the kidneys, the cocaine-induced metabolism of Cys was mainly anaerobic, with no change in the sulfate level (Figure 5G,H). Surprisingly, in the liver in rats self-administering cocaine, the sulfate levels increased significantly (Figure 4G) and then decreased after drug abstinence with extinction training (Figure 4H). In contrast, in the YC group, the sulfate levels in the presence of cocaine dropped significantly (Figure 4G), but returned to the control level after 10 days of drug abstinence with extinction training (Figure 4H). Such large differences in sulfate levels between the SA and YC groups receiving the same portions of cocaine at the same time indicate that the motivational mechanism of drug use affects the way the drug is metabolized in the liver. In support of this view, in a previous study, acute i.p. cocaine administration had no effect on sulfate levels while repeated i.p. cocaine administration increased their levels in the liver [17].

Increased levels of TSH, NPSH, and PSH were observed in the kidneys of rats self-administering cocaine (Figure 10A,C,E), indicating an increase in the reducing power. During drug abstinence with extinction training, in the SA group, the levels of TSH and PSH continued to be elevated (Figure 10B,F) while NPSH (mainly GSH) thiols decreased slightly (Figure 10D). In the rat liver, during cocaine self-administration, no changes in the contents of the examined thiol fractions (TSH, NPSH, and PSH) were observed (Figure 9A,C,E), while during drug abstinence with extinction training, decreases in the contents of TSH and PSH were noted both in SA and YC groups (Figure 9B,F).

As for GSH metabolizing enzymes, in the presence of cocaine, the activity of γ-GT (initiating the cleavage of GSH to glutamate and cysteinylglycine) was significantly increased in the liver of both SA and YC rats but was more pronounced in the YC group (Figure 11A). In general, the activity of this enzyme in the liver is relatively low, specifically many times lower than in the kidneys [35]. However, taking into account a high level of GSH in the liver, the increased activity of γ-GT in this tissue in the presence of cocaine could provide a way to increase free Cys concentration to replenish the plasma sulfane sulfur pool. Consistent with this reasoning, a much greater increase in γ-GT activity observed in the liver of YC rats after 10 days of drug abstinence with extinction training may be responsible for a very significant increase in the plasma sulfane sulfur pool, previously described in the YC group [16]. On the other hand, as a result of an increase in γ-GT activity in the liver of YC group (Figure 11A), the NPSH (mainly GSH) concentration decreased (Figure 9C) in the presence of cocaine but returned to the control level after 10 days of abstinence with extinction training (Figure 9D).

In the kidneys, unlike in the liver, γ-GT activity during cocaine intake decreased in the SA and YC group, (Figure 12A), presumably to limit the excessive availability of Cys for H_2_S production. On the other hand, after 10 days of cocaine abstinence with extinction training, the activity of γ-GT increased only in the SA group (Figure 12B) with a simultaneous decrease in NPSH content (Figure 10D) and no changes in the content of H_2_S (Figure 5F). However, in the YC group, H_2_S production was still increased (Figure 5F), and this effect was accompanied by an increased CSE activity (Figure 7B) while protein expression was maintained at the level of the YS control (Figure 8B). These differences in the activity of γ-GT in the SA and YC groups appear to be indicative of the motivational and pharmacological aspects of cocaine use.

Glutathione S-transferases (GSTs) are a family of enzymes originally involved in phase II of xenobiotic detoxification; however they, especially class pi (GSTP), can also act as glutathionylases in the S-glutathionylation reaction [43]. This process is a reversible oxidative modification on redox-sensitive protein Cys residues through the disulfide bond with GSH. S-glutathionylation is easily reversible via the release of GSH from Cys residues in the target proteins by glutaredoxin and thioredoxin [44]. This modification is considered as a defense mechanism to protect proteins against irreversible oxidative damage as well as is now recognized as an important process that regulates the function of target proteins [45]. Among many proteins susceptible to S-glutathionylation, two big categories include proteins involved in energy metabolism and signaling pathways [43]. A study by Uys et al. performed in the nucleus accumbens of cocaine-withdrawn rats before and after subsequent acute cocaine injection suggested an association between S-glutathionylation and GST activity [46]. A recent study by Kruyer et al. also suggests an important role of S-glutathionylation of some proteins during cocaine seeking [47].

In our study, we observed an increased GST activity in the liver of the SA and YC groups during cocaine intake (Figure 11C) but, after 10 days of abstinence with extinction training, it decreased significantly in the YC group while, in the SA one, there was only a decreasing trend (Figure 11D). Interestingly, the level of PSH groups (which can be a measure of protein Cys modification) was unchanged at the presence of cocaine in the liver of the SA and YC groups (Figure 9E) but was significantly decreased during drug abstinence with extinction training in both groups (Figure 9F). It can be supposed that, in our study, GST activation during cocaine intake leads to an increase in the S-glutathionylation process found after 10 days of drug abstinence despite the observed decreasing trend in GST activity in the SA group and a significant decline in its activity in the YC group (Figure 11D). A similar effect was described by Tew and Townsend in rat nucleus accumbens, where withdrawal from daily cocaine followed by acute saline reduced GSTP expression and increased S-glutathionylation [43]. In contrast to the liver, in the kidney of the YC rats, a significant decrease in the GST activity was found during cocaine intake (Figure 12C) while, after 10 days of drug abstinence with extinction training, only a downward trend was observed (Figure 12D). In turn, the level of PSH groups during drug abstinence with extinction training was increased in both SA and YC groups, which can suggest a decrease in the S-glutathionylation process.

## 5. Conclusions

Our study shows that cocaine in the self-administration model and in abstinence with extinction training affects thiol homeostasis and RSS formation in the peripheral tissues. We hypothesize that modifications of Cys residues of some target proteins in the studied tissues by persulfidation and S-glutathionylation play an important role in both the pharmacological and motivational effects of cocaine. Both these oxidative processes are reversible and can be regarded as a mechanism which may alter protein function and may transduce a signal into a functional response. Our study suggests that, under the influence of cocaine, S-glutathionylation processes dominate in the liver while, in the kidney, the signal is transduced mainly via persulfidation of some proteins. Cocaine influences the regulation of enzymes involved in the synthesis and transport of RSS, both at the level of expression and activity of these proteins. The obtained results imply also that thiol redox regulation plays an important role in the motivational and pharmacological effects of cocaine.

## Figures and Tables

**Figure 1 antioxidants-10-00074-f001:**
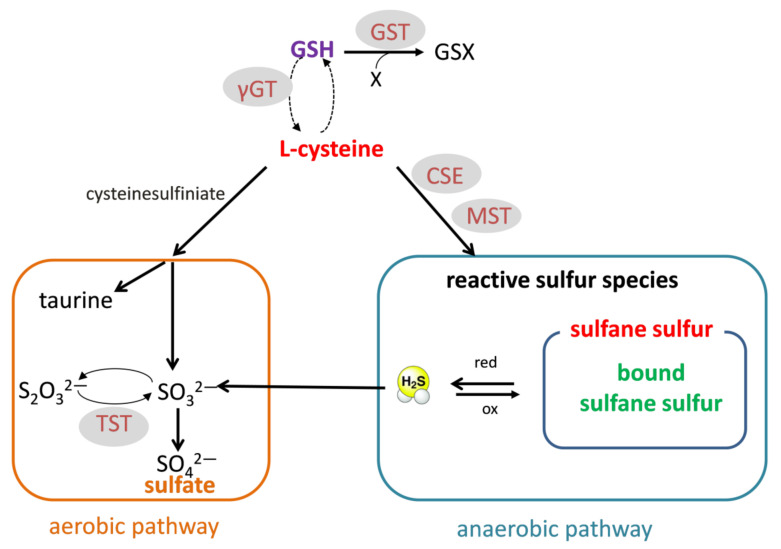
l-cysteine as a substrate for glutathione (GSH) synthesis as well as a source of reactive sulfur species (RSS), including sulfane sulfur, bound sulfane sulfur, and H_2_S produced during anaerobic metabolism. l-cysteine can also be metabolized by the aerobic pathway to sulfate or taurine. GST—glutathione S-transferase; γ-GT—gamma glutamyl transpeptidase; CSE—cystathionine γ-lyase; MST—3-mercaptopyruvate sulfurtransferase; TST—rhodanese.

**Figure 2 antioxidants-10-00074-f002:**
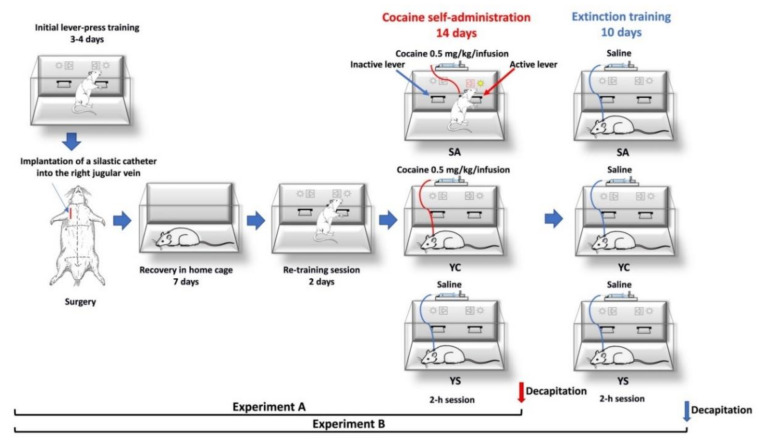
Graphical diagram illustrating the successive steps of the experimental procedure. SA—cocaine self-administration, YC—yoked cocaine administration, YS—yoked saline administration.

**Figure 3 antioxidants-10-00074-f003:**
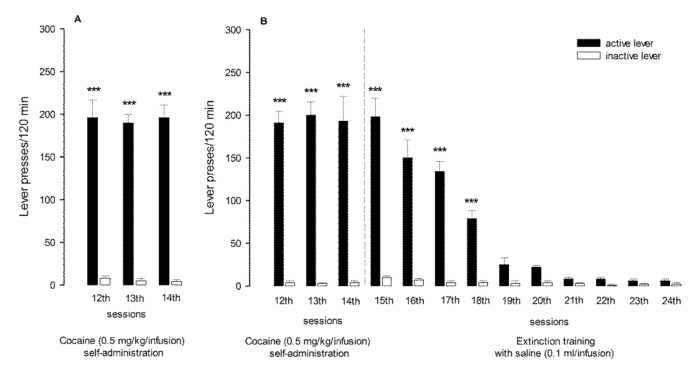
The number of active and inactive lever presses during the last 3 days of cocaine self-administration (**A**) as well as during the last 3 days of cocaine self-administration and following 10 days of abstinence with extinction training (**B**): the number of rats per group, *n* = 10. Data are presented as the mean ± SEM, *** *p* < 0.001 vs. inactive lever.

**Figure 4 antioxidants-10-00074-f004:**
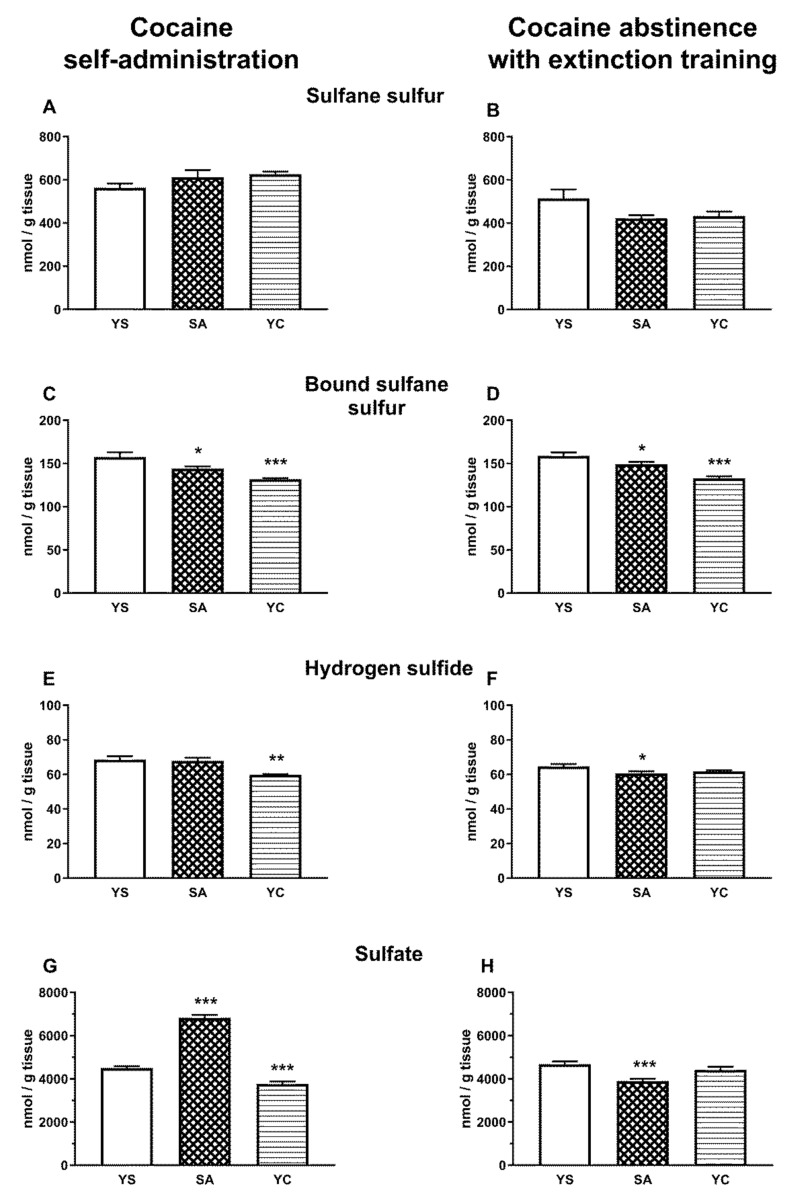
Concentrations of the total sulfane sulfur, bound sulfane sulfur, hydrogen sulfide, and sulfate in the presence of cocaine (**A**,**C**,**E**,**G**) and after 10 days of drug abstinence with extinction training (**B**,**D**,**F**,**H**) in the livers of rats self-administering cocaine (SA), receiving passive cocaine infusions (YC), or receiving saline infusions (YS): data are presented as the mean ± SEM, *n* = 8–10 per group. * *p* < 0.05, ** *p* < 0.01, *** *p* < 0.001 vs. YS group.

**Figure 5 antioxidants-10-00074-f005:**
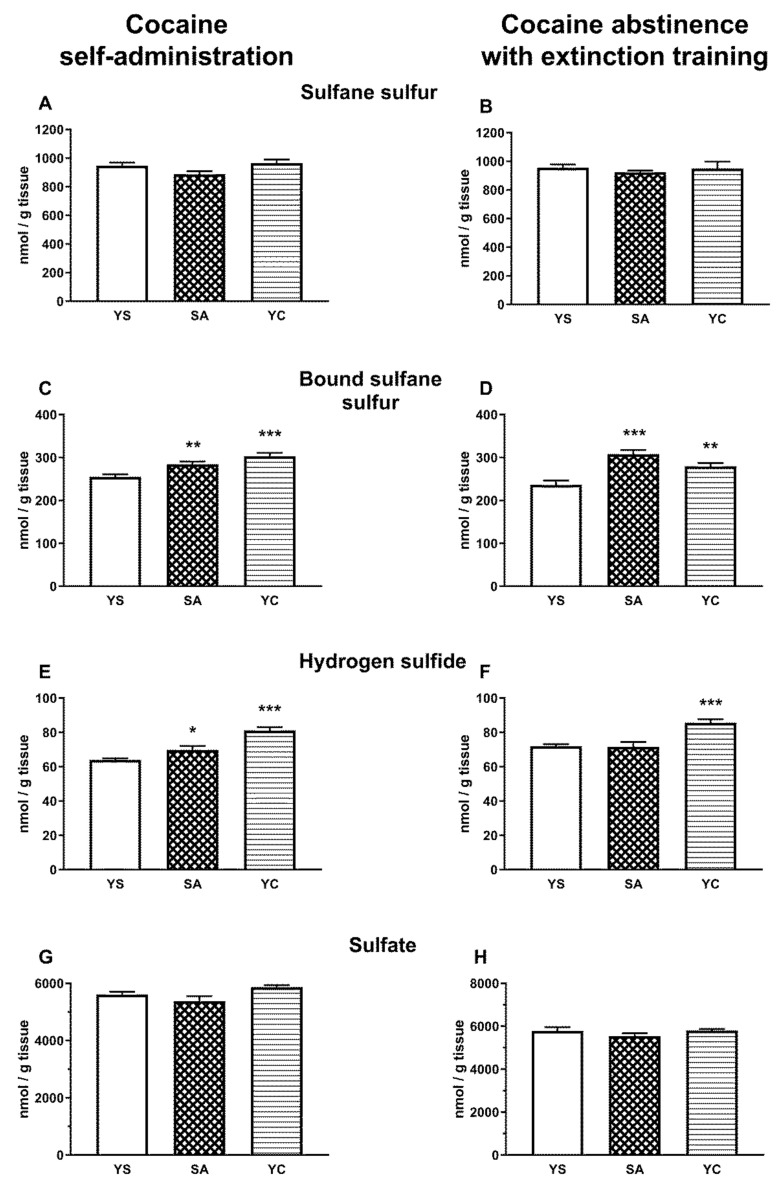
Concentrations of the total sulfane sulfur, bound sulfane sulfur, hydrogen sulfide, and sulfate in the presence of cocaine (**A**,**C**,**E**,**G**) and after 10 days of drug abstinence with extinction training (**B**,**D**,**F**,**H**) in the kidneys of rats self-administering cocaine (SA), receiving passive cocaine infusions (YC), and receiving saline infusions (YS): data are presented as the mean ± SEM, *n* = 8–10 per group. * *p* < 0.05, ** *p* < 0.01, *** *p* < 0.001 vs. YS group.

**Figure 6 antioxidants-10-00074-f006:**
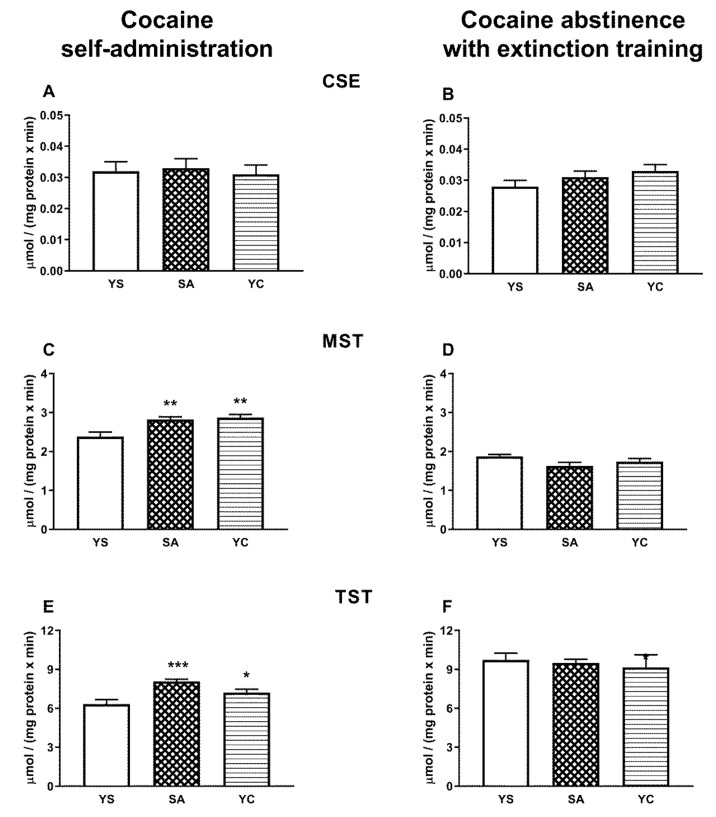
The activity of CSE, MST, and TST in the presence of cocaine (**A**,**C**,**E**) and after 10 days of drug abstinence with extinction training (**B**,**D**,**F**) in the livers of rats self-administering cocaine (SA), receiving passive cocaine infusions (YC), and receiving saline infusions (YS): data are presented as the mean ± SEM, *n* = 8–10 per group. * *p* < 0.05, ** *p* < 0.01, *** *p* < 0.001 vs. YS group.

**Figure 7 antioxidants-10-00074-f007:**
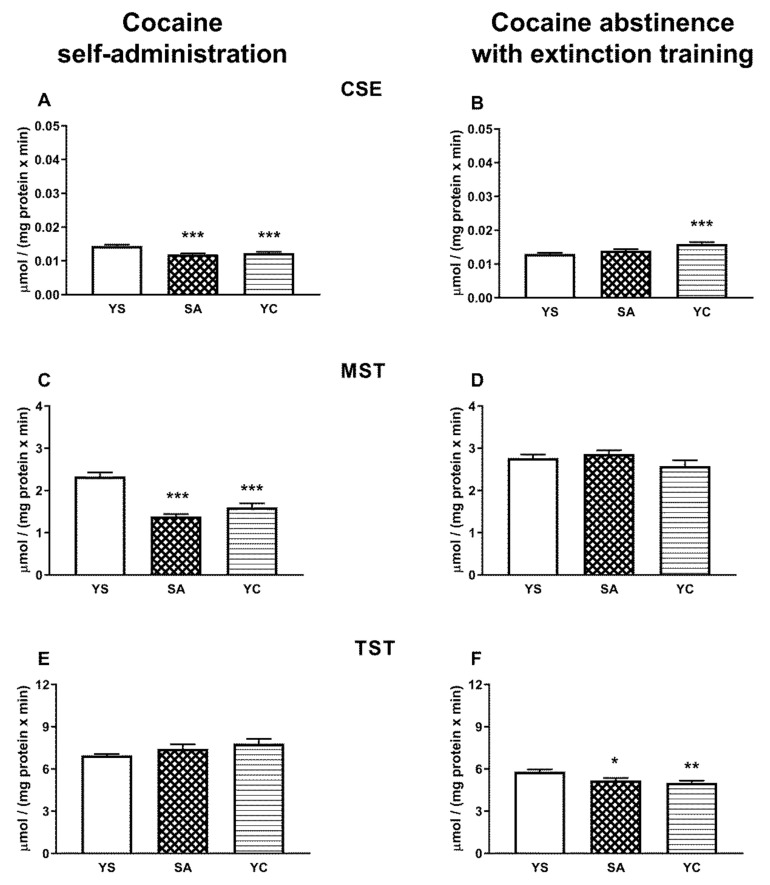
The activity of CSE, MST, and TST in the presence of cocaine (**A**,**C**,**E**) and after 10 days of drug abstinence with extinction training (**B**,**D**,**F**) in the kidneys of rats self-administering cocaine (SA), receiving passive cocaine infusion (YC), and receiving saline infusions (YS): data are presented as the mean ± SEM, *n* = 8–10 per group. * *p* < 0.05, ** *p* < 0.01, *** *p* < 0.001 vs. YS group.

**Figure 8 antioxidants-10-00074-f008:**
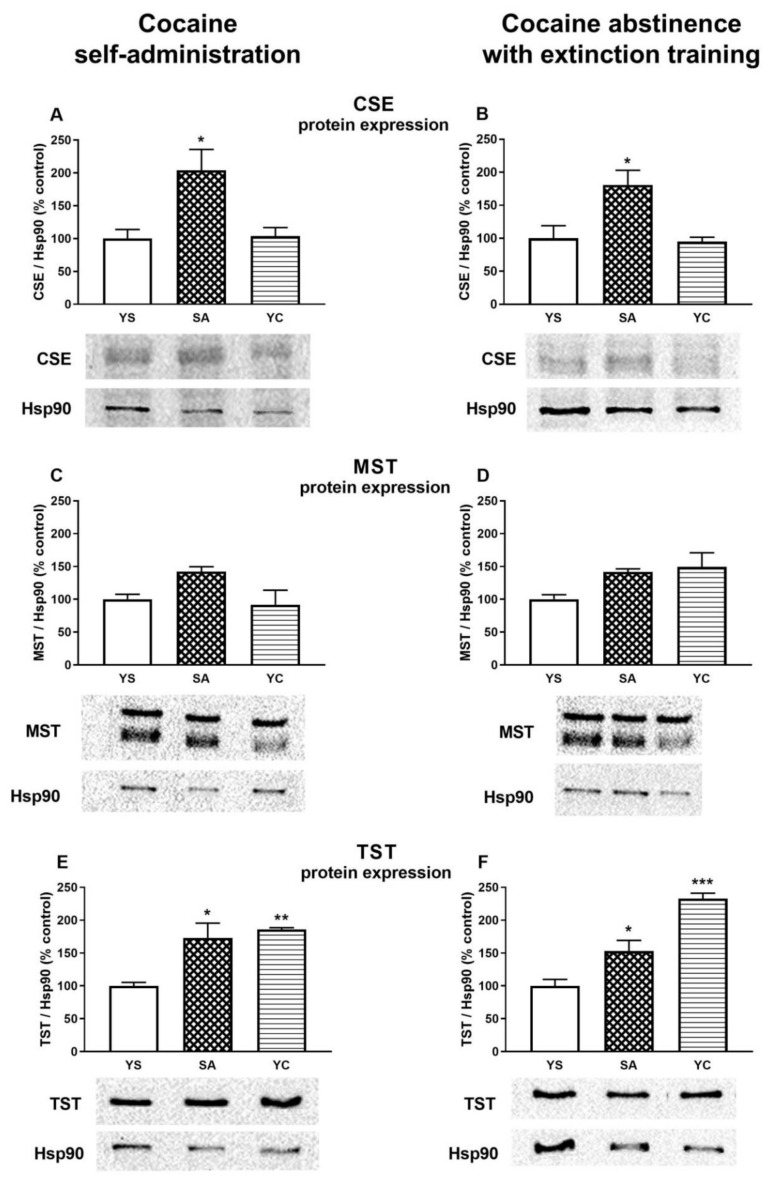
Western blot analysis of the CSE, MST, and TST protein expression in the presence of cocaine (**A**,**C**,**E**) and after 10 days of drug abstinence with extinction training (**B**,**D**,**F**) in the kidneys of rats self-administering cocaine (SA), receiving passive cocaine infusion (YC), and receiving saline infusions (YS): data are presented as the mean ± SEM, *n* = 3 rats per each group. * *p* < 0.05, ** *p* < 0.01, *** *p* < 0.001 vs. YS group. One representative band for the studied group (YS, SA, and YC) is shown under each graph.

**Figure 9 antioxidants-10-00074-f009:**
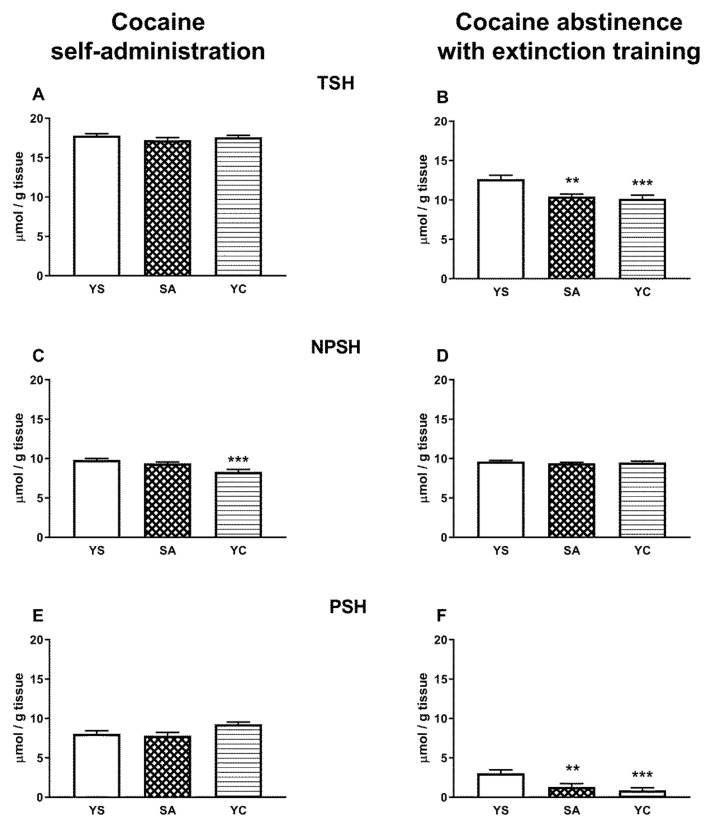
Levels of the total, non-protein, and protein –SH groups in the presence of cocaine (**A**,**C**,**E**) and after 10 days of drug abstinence with extinction training (**B**,**D**,**F**) in the livers of rats self-administering cocaine (SA), receiving passive cocaine infusions (YC), and receiving saline infusions (YS): data are presented as the mean ± SEM, *n* = 8–10 per group. ** *p* < 0.01, *** *p* < 0.001 vs. YS group.

**Figure 10 antioxidants-10-00074-f010:**
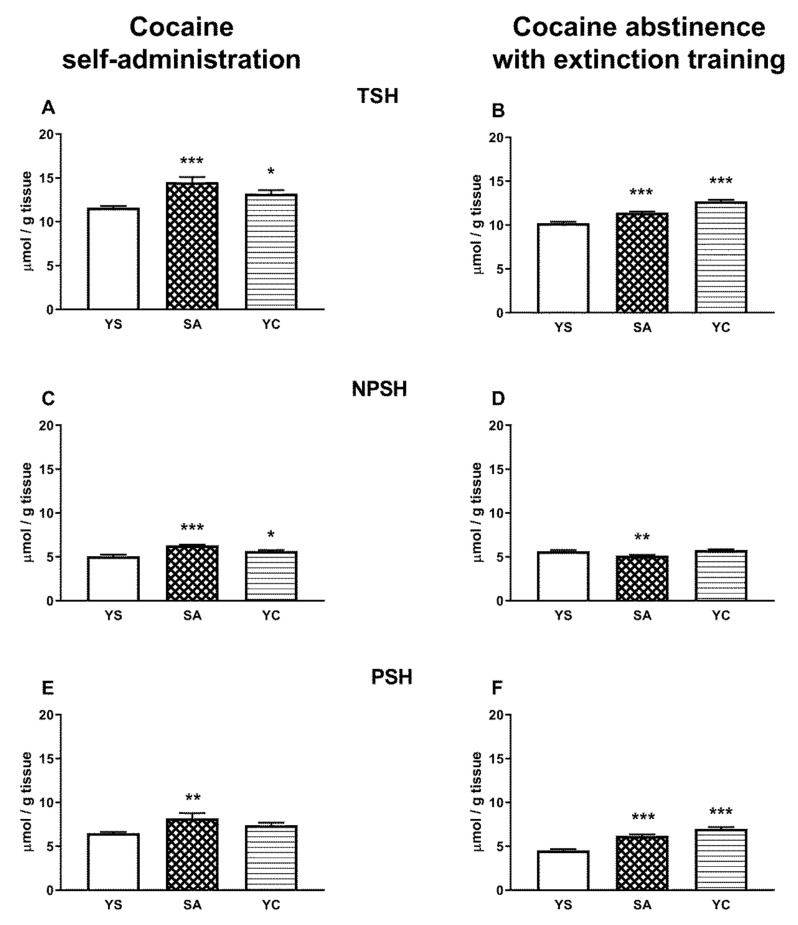
Levels of the total, non-protein, and protein –SH groups in the presence of cocaine (**A**,**C**,**E**) and after 10 days of drug abstinence with extinction training (**B**,**D**,**F**) in the kidneys of rats self-administering cocaine (SA), receiving passive cocaine infusions (YC), and receiving saline infusions (YS): data are presented as the mean ± SEM, *n* = 8–10 per group. * *p* < 0.05, ** *p* < 0.01, *** *p* < 0.001 vs. YS group.

**Figure 11 antioxidants-10-00074-f011:**
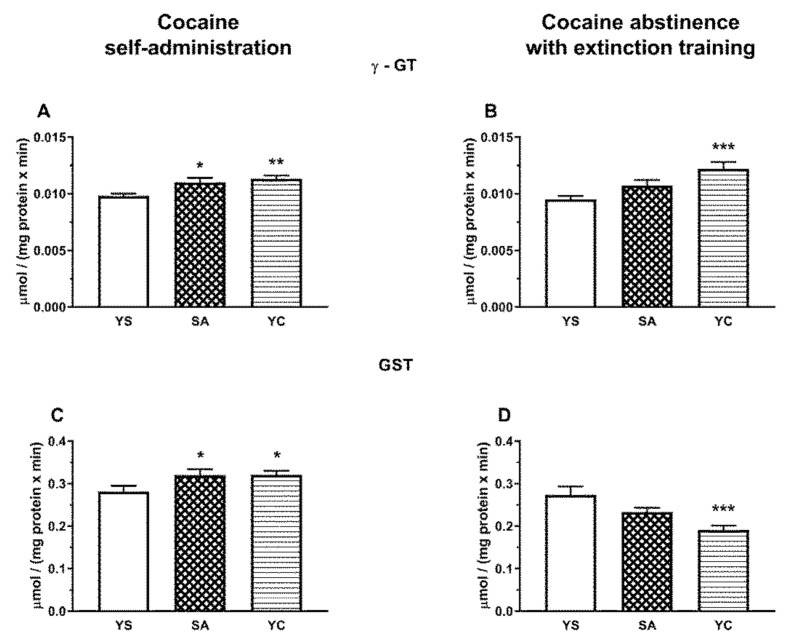
The activity of γ-GT and GST in the presence of cocaine (**A**,**C**) and after 10 days of drug abstinence with extinction training (**B**,**D**) in the livers of rats self-administering cocaine (SA), receiving passive cocaine infusions (YC), and receiving saline infusions (YS). Data are presented as the mean ± SEM, *n* = 8–10 per group. * *p* < 0.05, ** *p* < 0.01, *** *p* < 0.001 vs. YS group.

**Figure 12 antioxidants-10-00074-f012:**
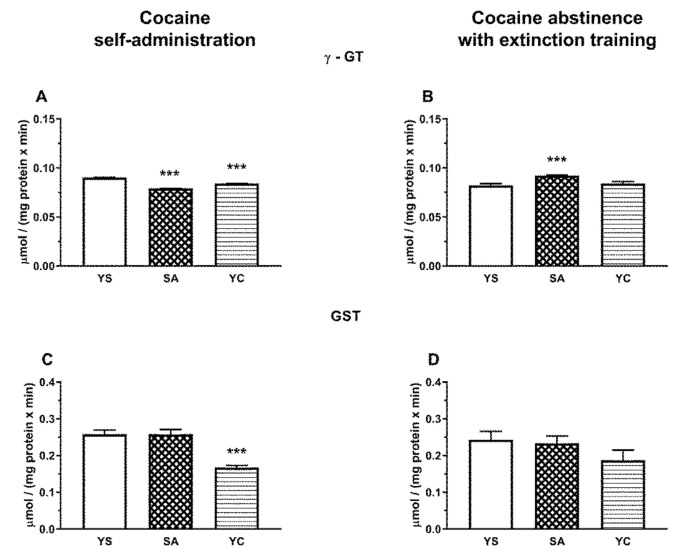
The activity of γ-GT and GST in the presence of cocaine (**A**,**C**) and after 10 days of drug abstinence with extinction training (**B**,**D**) in the kidneys of rats self-administering cocaine (SA), receiving passive cocaine infusions (YC), and receiving saline infusions (YS): data are presented as the mean ± SEM, *n* = 8–10 per group. *** *p* < 0.001 vs. YS group.

## Data Availability

The data presented in this study are available on request from the corresponding author. The data are not publicly available due to reasons of privacy.

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
