# Peer review of "Evaluation of Cysteine Metabolism in the Rat Liver and Kidney Following Intravenous Cocaine Administration and Abstinence"

_antioxidants, 2021, doi:10.3390/antiox10010074_

Round 1

Reviewer 1 Report

The manuscript entitled “Evaluation of cysteine metabolism in the rat liver and kidney following intravenous cocaine administration and abstinence period” by Kowalczyk-Pachel et al investigates production of reactive sulphur species in the liver and kidney of rats in response to self-administration of cocaine. The study compares production of metabolites and activity of enzymes which generate such metabolites between passive and self-administered groups and compare cocaine self-administered groups with those experiencing abstinence.  The paper is generally well written and easy to follow and experiments appear to have been well planned and data clearly presented.  There are however issues which should be addressed experimentally before the paper be considered for publication.

  1. The authors use biochemical assays to determine which enzymes may be responsible for the changes metabolites observed in (Figs 2, 3). The authors then go to use further assays to assess the activity of these enzymes in tissue (Fig 5, 6, 9 and 10) and report significant increases and decreases in values between groups. It would be extremely helpful if the authors would make clear if this is due to a change in the quantity of enzyme present per unit of tissue following treatment or an increase in the activity of the enzyme per se.  This should be demonstrated in a series of western blots using antibodies specific to the enzymes under investigation.
  2. The introduction is too long and should clearly state the rationale behind the study and clearly state hypotheses which are being tested here.
  3. The discussion is far too long. This section should pull the main points of the manuscript into focus and offer some perspective on the key questions and future directions of this area of work.

Author Response

Reviewer 1:

Comments and Suggestions for Authors

The manuscript entitled “Evaluation of cysteine metabolism in the rat liver and kidney following intravenous cocaine administration and abstinence period” by Kowalczyk-Pachel et al investigates production of reactive sulphur species in the liver and kidney of rats in response to self-administration of cocaine. The study compares production of metabolites and activity of enzymes which generate such metabolites between passive and self-administered groups and compare cocaine self-administered groups with those experiencing abstinence. The paper is generally well written and easy to follow and experiments appear to have been well planned and data clearly presented. There are however issues which should be addressed experimentally before the paper be considered for publication.

The authors use biochemical assays to determine which enzymes may be responsible for the changes metabolites observed in (Figs 2, 3).The authors then go to use further assays to assess the activity of these enzymes in tissue (Fig 5, 6, 9 and 10) and report significant increases and decreases in values between groups. It would be extremely helpful if the authors would make clear if this is due to a change in the quantity of enzyme present per unit of tissue following treatment or an increase in the activity of the enzyme per se. This should be demonstrated in a series of western blots using antibodies specific to the enzymes under investigation.

Response to above suggestion

In order to answer the question of whether the changes in the activity of the tested enzymes are related to a decrease or increase in the level of proteins of the tested enzymes, we performed an additional preliminary Western blot analysis for 3 enzymes: CSE, MST and TST only in the kidneys of rats due to the fact that the most pronounced changes in the levels of RSS (bound sulfane sulfur and H2S) were observed in this tissue. In the future, we plan to develop further research in this direction to explain in detail the role of sulfurtransferases in the cocaine mode of action.

The obtained results were added to the current version of the manuscript (methods - page 6, lines 213-229, results - page 12, lines327-339, Figure 8, page 13-14, lines 341-347).

Reviewer's suggestion

The introduction is too long and should clearly state the rationale behind the study and clearly state hypotheses which are being tested here. The discussion is far too long. This section should pull the main points of the manuscript into focus and offer some perspective on the key questions and future directions of this area of work.

Response to above suggestion

We have invested much effort in shortening the introduction. The changed fragments of the manuscript are highlighted in yellow.

Reviewer 2 Report

This study presents the effect of intravenous cocaine administration and abstinence period on thiols and RSS in the rat liver and kidney. The data are potentially interesting and worthy of eventual publication. I think however that there are a few improvements that should be made before publication. The authors should also clarify/correct the points listed below.

  • The animal experimental procedure 2.2 and 2.3. could be clearly illustrated instead of being described only in the text – this would make it easy to understanding.

  • Please spell out CNS stimulation.

I hope these comments will be helpful.

Author Response

Reviewer 2

Comments and Suggestions for Authors

This study presents the effect of intravenous cocaine administration and abstinence period on thiols and RSS in the rat liver and kidney. The data are potentially interesting and worthy of eventual publication. I think however that there are a few improvements that should be made before publication. The authors should also clarify/correct the points listed below.

The animal experimental procedure 2.2 and 2.3. could be clearly illustrated instead of being described only in the text – this would make it easy to understanding.

Please spell out CNS stimulation

I hope these comments will be helpful.

Response to above suggestion

As suggested by the reviewer, the experimental procedure is presented graphically in Figure 2. The attached description of the course of the entire experiment was slightly modified and adapted to Figure 2. Due to the addition of this drawing, the numbering of the remaining figures throughout the text has changed.

Abbreviation CNS has been replaced with its full name (page 1, line 36).

Round 2

Reviewer 1 Report

Kowalczyk-Pachel et al present a revised version of the manuscript entitled "Evaluation of cysteine metabolism in the rat liver and kidney following intravenous cocaine administration and abstinence period".  

The authors present a panel of western blots which assess the levels of the enzymes in question present in kidney tissue of treated rats.  This issue appears to be resolved and the authors should be congratulated for producing a series of good quality blots.  

Minor

The introduction and discussion sections, although modified, are still in the referees view, far too long.